# The Anti-Dengue Virus Peptide DV2 Inhibits Zika Virus Both In Vitro and In Vivo

**DOI:** 10.3390/v15040839

**Published:** 2023-03-25

**Authors:** Maria Fernanda de Castro-Amarante, Samuel Santos Pereira, Lennon Ramos Pereira, Lucas Souza Santos, Alexia Adrianne Venceslau-Carvalho, Eduardo Gimenes Martins, Andrea Balan, Luís Carlos de Souza Ferreira

**Affiliations:** 1Laboratory of Vaccine Development, Institute of Biomedical Sciences, University of São Paulo, São Paulo 05508-000, Brazil; 2Scientific Platform Pasteur USP, São Paulo 05508-020, Brazil; 3Applied Structural Biology Laboratory, Institute of Biomedical Sciences, University of São Paulo, São Paulo 05508-000, Brazil

**Keywords:** Zika virus, peptide, antiviral, DV2 peptide, dengue virus, flavivirus

## Abstract

The C-terminal portion of the E protein, known as stem, is conserved among flaviviruses and is an important target to peptide-based antiviral strategies. Since the dengue (DENV) and Zika (ZIKV) viruses share sequences in the stem region, in this study we evaluated the cross-inhibition of ZIKV by the stem-based DV2 peptide (419–447), which was previously described to inhibit all DENV serotypes. Thus, the anti-ZIKV effects induced by treatments with the DV2 peptide were tested in both in vitro and in vivo conditions. Molecular modeling approaches have demonstrated that the DV2 peptide interacts with amino acid residues exposed on the surface of pre- and postfusion forms of the ZIKA envelope (E) protein. The peptide did not have any significant cytotoxic effects on eukaryotic cells but efficiently inhibited ZIKV infectivity in cultivated Vero cells. In addition, the DV2 peptide reduced morbidity and mortality in mice subjected to lethal challenges with a ZIKV strain isolated in Brazil. Taken together, the present results support the therapeutic potential of the DV2 peptide against ZIKV infections and open perspectives for the development and clinical testing of anti-flavivirus treatments based on synthetic stem-based peptides.

## 1. Introduction

The Zika (ZIKV) and dengue (DENV) viruses belong to the Flaviviridae family, which also includes other clinically important viruses, such as yellow fever virus (YFV), West Nile virus (WNV), and Japanese encephalitis virus (JEV) [1,2]. DENV infection in humans has been detected in more than 100 countries, for which observers estimate about 390 million cases and 22,000 deaths by dengue disease worldwide each year [3,4]. On the other hand, around 1.5 million of ZIKV infections have been reported in more than 70 countries [5]. The Zika virus was probably introduced into Brazil in 2013 [6], where two years later it was related to the occurrence of neurologic damage in newborn babies [7]; it was and declared a global health emergency by the World Health Organization (WHO) in 2016 [8]. Even though ZIKV infections are usually associated with self-limiting symptoms, severe clinical outcomes may be observed, such as Congenital Zika Syndrome (CZS) and Guillain-Barré syndrome [9]. In Brazil, between 2015 and 2022, approximately 3707 cases of congenital ZIKV infection were confirmed, of which 1852 were classified as CZS. Additionally, 31,194 cases of severe dengue were reported in the period from 2020 to 2022 [10]. Thus, the high incidence of these infections over the years and the related clinical complications illustrate the importance and impact of these diseases on public health.

Several DENV and ZIKV vaccine candidates have been developed and tested under preclinical and clinical conditions [11,12]. Two tetravalent dengue vaccines have already been licensed, Dengvaxia^®^ (Sanofi Pasteur) and QDENGA^®^ (Takeda), which are based on chimaeric live-attenuated viruses [13]. Moreover, part of the anti-ZIKV vaccine strategies have evolved into phase I and II clinical trials (Clinical trials: NCT03008122, NCT03014089, NCT03110770, NCT02996461). The reported results demonstrated that formulations based on inactivated virus, DNA vaccine, and more recently, lipid nanoparticle-encapsulated mRNA-based vaccine encoding the prM-E antigens were well-tolerated and immunogenic, with induction of neutralizing antibodies in immunized individuals [14,15,16,17,18,19]. However, to date, none of the anti-ZIKV vaccines have been approved for use in humans. On the other hand, effective therapeutic strategies may reduce the lethality and morbidity related to severe forms of DENV and ZIKV infections. In this sense, different natural or synthetically produced antiviral candidates have also been tested against those arboviruses, including bioactive compounds from plant extracts, small molecules, or peptides [20,21,22,23,24,25,26]. Besides the high number of studies regarding the discovery of new compounds that have antiviral activity against those viruses, only a few have been tested in humans. Thus, so far, there are no specific drugs with proven efficacy against DENV, ZIKV, or other flaviviruses, and currently available therapies are only palliative [27].

Previous studies have demonstrated that synthetic peptides (<100 amino acid residues) have antiviral effects, and some of them show features compatible with pharmacological uses [28]. Furthermore, studies have demonstrated the inhibitory effects of synthetic peptides against different flaviviruses, such as DENV [29,30,31,32,33], JEV [34], WNV [35], and more recently, ZIKV [36]. Peptides derived from the proximal stem region of the DENV2 E protein inhibited viral infectivity under in vitro conditions. In addition, detailed analyses demonstrated that DENV2-inhibitory peptides act via a two-step mechanism. First, the peptides bind nonspecifically to the virion membrane at neutral pH using C-terminal hydrophobic residues; subsequently, at low pH into the endosome, the inhibitory peptides interact with the postfusion form of the E protein, blocking the membrane fusion step and, thus, the virus’ infectivity. Specifically, a synthetic peptide named DV2, which was designed according to the domain III (EDIII) of the stem region of the E protein amino acid sequence, targets a late step of the viral fusion process and has been reported to inhibit the infectivity of all DENV serotypes [32,33]. Nonetheless, there is no additional data regarding the activity of the DV2 peptide on other flaviviruses, as well as on its protective effects in vivo under experimental conditions.

In the present study, we evaluated the interactions of the DV2 peptide with the ZIKV E protein using in silico analyses and biological effects data. Experimental data demonstrated that the DV2 peptide blocks ZIKV infectivity both in cultivated cells and under in vivo conditions, as demonstrated in immunodeficient mice subjected to lethal challenges with the virus, without any detected cytotoxic side effects.

## 2. Materials and Methods

### 2.1. Ethics Statement

The protocols were approved on 28 March 2016 by the Institutional Animal Care and Use Committee (CEUA) of the University of São Paulo (protocol number 22/2016) and were conducted according to the Ethical Principles of Animal Experimentation established by the Brazilian College of Animal Experimentation.

### 2.2. Cell Lines and Virus

The Vero CCL-81 cell line was cultured in DMEM supplemented with 10% fetal bovine serum (FBS) (Life Technologies). Aedes albopictus C6/36 cells were cultured in Leibovitz L-15 medium (Vitrocell, Campinas, SP, Brazil) supplemented with 2% FBS. ZIKV^BR^ strain (isolated from a clinical case during the 2015 Brazilian outbreak (isolate = “BeH823339”, GenBank: KU729217) [37] was propagated in C6/36 cells, concentrated, and titrated as previously described [38,39].

### 2.3. Peptide Synthesis

DV2 (AWDFGSLGGVFTSIGKALHQVFGAIYGAA) and Ph-22 (ACNTIDPRHCGGGSAETVES)—a synthetic peptide previously generated in the laboratory with no reported activity against ZIKV—were purchased from Biomatik (Piscataway, NJ, USA), solubilized in dimethylsulfoxide (DMSO) to obtain stock solutions with end concentrations of 2.5 mM and 5 mM, respectively, and stored at −20 °C until use. The purity of the peptides was higher than 95%.

### 2.4. Cytotoxicity Tests

The cytotoxic effects of the DV2 peptide were evaluated by measuring cell metabolic activity as well as cell viability. Cell proliferation was assessed using the reagent WST-1 (Roche) according to the manufacturer’s instructions, and the number of live cells was determined by flow cytometry. Briefly, Vero CCL-81 cell monolayers established in 96-well plates were incubated at 37 °C for 24 h with increasing concentrations of the peptide ranging from 0.75 to 24 μM. Equivalent volumes of DMSO were used as the control. The WST-1 reagent was added to the wells, which were then incubated for 1 h at 37 °C. The absorbance was measured using a microplate reader at 440 nm. For flow cytometry analysis, live/dead aqua fluorescent reactive dye (Invitrogen) was added to the wells after a washing step for incubation for 30 min. at room temperature. The cells were then acquired on a LSR FortessaTM cytometer (BD, Franklin Lakes, NJ, USA). The data were analyzed using FlowJo software (version 10, Tree Star, San Carlo, CA, USA).

### 2.5. In Silico Docking of the DV2 Peptide

Prediction of the DV2 peptide-binding site on the ZIKV E protein in the postfusion conformation was performed using a three-dimensional model built with Modeller v10.1 [40] based on the structural coordinates from homologous DENV1 (PDB:3G7T) and DENV2 (PDB:1OK8). A total of 100 models were generated, and the selected structure was chosen on the basis of the DOPE score. Peptide docking was performed using CABS-dock v0.9.18 [41] with both conformations of the ZIKV E protein as docking pairings. Each simulation generated 10,000 models (from 10 independent trajectories), which were further reduced to 1000 models on the basis of low-energy selection. The 1000 top-scored models were then classified into 10 clusters, and the all-atom structures were derived by PD2 [42] from each cluster’s medoid. The Zika E protein residues involved in the interaction with the DV2 peptide were identified using the Protein-Ligand Interaction Profiler (PLIP) server [43]. The PyMOL molecular graphics system (DeLano Scientific, San Carlos, CA, USA; DeLano 2002) was used for analyzing the structures and interactions of DV2 and the ZIKV E protein and for preparing the figures.

### 2.6. Virus Inhibition Assays

Different concentrations (0.75, 1.5, 3.0, 6.0, and 12 µM) of the DV2 or Ph22 peptides were incubated with ZIKV^BR^ strain (MOI of 1) for 30 min at 37 °C and 5% CO2. The virus-peptide mixture was transferred to Vero CCL-81 cell monolayers (previously established in 96-flat well plates) and incubated for 1 h at 37 °C and 5% CO2. After washing (three times) with PBS, fresh DMEM supplemented with 2% FBS was added to the cells and incubated for up to 24 h. The cell culture supernatants were harvested and stored at −80 °C. The virus titers of the cell-free supernatants were determined by plaque assays (PFU/mL), as previously described [38].

### 2.7. Flow Cytometry Analyses of ZIKV-Infected Cells

The cell monolayers were washed twice with PBS and trypsinized using 1× Trypsin/EDTA (Gibco). Cells were fixed/permeabilized using a Cytofix/Cytoperm kit (BD Bioscience), according to the manufacturer’s instructions, labeled with the 4G2 monoclonal antibody (Millipore) (10 µg/mL), and incubated with a rabbit anti-mouse IgG antibody coupled to AF488 (Thermo Fisher Scientific) (final dilution of 1:1000). Flow cytometry analyses were performed using an LSR FortessaTM cytometer (BD, Franklin Lakes, NJ, USA). The data were analyzed using FlowJo software (version 10, Tree Star, San Carlo, CA, USA) to determine the percentage of infected cells.

### 2.8. Mice Infection

The AG129 (IFNα/βR^−−^) (7–8 weeks old) mice used in this study were bred under specific pathogen-free conditions at the Isogenic Mouse Facility of the Microbiology Department, University of São Paulo, Brazil. ZIKV^BR^ preparations (100 PFU/animal) were incubated at 37 °C for 30 min. with 12 µM DV2 peptide (*n* = 5) or equivalent volumes of DMSO (*n* = 5), maintaining a final peptide-virus mixture volume of 50 µL injected per animal. Mice inoculated with ZIKV only (*n* = 4) and naive animals (*n* = 3) were used as the control groups. Naïve AG129 mice were s.c. inoculated with the peptide/DMSO-virus mixtures and monitored for up to 17 days. Morbidity signs were quantified on the basis of an arbitrary score scale (healthy, score 0; ruffled fur, score 1; paralysis, score 2; deformed spinal column, score 3; moribund, score 4) together with body weight measurements. Serum samples were collected on days 3, 5, 7, and 14 after infection and stored at −80 °C for analysis of viremia. A body weight reduction of 20% was used as a parameter for sacrificing animals in combination with morbidity evaluation (clinical score of 4).

### 2.9. Statistical Analysis

Statistical analyses were performed using Prism 6 (GraphPad Software Inc., LA Jolla, CA, USA). A *t*-test was used for two-group comparisons, whereas two-way ANOVA followed by Bonferroni correction was used when the data involved several groups and more than one variable (concentrations). The log-rank test (Mantel-Cox) was used to analyze the survival and morbidity data. Differences were considered significant when the *p*-value (*p*) was ≤0.05.

## 3. Results

### 3.1. In-Silico Docking Analyses of DV2 with the ZIKV E Protein

In silico docking analyses of the DV2 peptide were carried out with the postfusion conformation of the ZIKV E protein-binding site using the CABS-dock program. We used a structural model based on the coordinates of DENV1 and DENV2 proteins as the input for the program. The best docking complex was selected among those with the lowest energy parameters; it corresponded to the positioning of the DV2 peptide on a protein surface rich in hydrophobic and negatively charged amino acids, interacting with domains I and II (Figure 1). According to the docking simulation, the interaction of the peptide in the postfusion conformation mainly relies on residues from the conserved region (419–441) and involves hydrogen bonds, hydrophobic interactions, and salt bridges. These data are in accordance with previous results and models that describe the interaction of peptides derived from the membrane-proximal region of the dengue virus E protein in the postfusion conformation [32]. According to the PLIP program, 11 residues of the DV2 peptide participate in the interactions with ZIKV E protein, 8 in the conserved region, and only 3 among residues 441–447, reinforcing the notion that the conserved region is important for binding in the trimeric form. A list of the predicted interactions for all the residues is shown in Appendix A (Appendix A).

### 3.2. In Vitro Cytotoxicity and Anti-ZIKV Effects of the DV2 Peptide

Based on the expected interactions among the amino acid residues of DV2 and ZIKV E proteins, we set up experiments to evaluate the in vitro effects of the peptide on ZIKV-infected cells. Initially, we determined the cytotoxic effects of DV2 on Vero CCL-81 cells incubated with the peptide for up to 24 h. The DV2 citotoxicity was evaluated measuring the number of live cells by staining them with a live/dead fluorescent and determining the cell metabolic activity using the WST-1 reagent. As shown in Figure 2, the DV2 peptide did not show any significant cytotoxicity at concentrations ranging from 0.7 to 24 µM. Similarly, the control (DMSO) tested at equivalent dilutions did not induce any significant cell death.

Next, we tested the in vitro antiviral effects of the DV2 peptide. Vero cells were infected with ZIKV (MOI of 1) in the presence of either the DV2 or the control Ph22 peptides, followed by detection of infected cells and infectious viral particles in the cell supernatants collected 24 h post infection (hpi). Notably, the DV2 peptide drastically reduced ZIKV infectivity at concentrations ranging from 3 to 12 µM, as measured by the number of infected cells (Figure 3a). The anti-ZIKV effects of DV2, compared to that of the control Ph22 peptide, reached 80% (3 μM) to 97.5% (12 μM) inhibition. Similarly, determination of the viable viral particles in the culture supernatants of the infected cells demonstrated that concentrations of 6 and 12 μM of DV2 reduced the number of PFU to values below the detection limit of the assay (Figure 3b). The inhibitory concentrations (IC50 and IC90) are listed in Table 1 and Appendix A (Appendix A).

### 3.3. Protective Effects of the DV2 Peptide in AG129 Mice Challenged with ZIKV

The in vivo anti-ZIKV effects of the DV2 peptide were determined in AG129 IFNα/βR^−^/^−^ mice following s.c. inoculation with a lethal dose of ZIKV^BR^. As demonstrated in Figure 4, incubation of ZIKV with the DV2 peptide reduced morbidity signs in infected animals (Figure 4a–e). DV2 also prevented body weight loss and viremia (Figure 4f,g). Additionally, we measured the protection conferred by the DV2 peptide against mortality in ZIKV-infected mice. In contrast to mice inoculated with ZIKV particles in the presence of DMSO, 80% of mice infected with ZIKV exposed to 12 µM DV2 peptide survived the lethal challenge (Figure 4h). Taken together, these results support a strong anti-ZIKV effect of the DV2 peptide both in vitro and in vivo.

## 4. Discussion

The flavivirus E protein is a surface membrane protein involved in host cell receptor-binding functions and mediates the fusion of the viral envelope with endosomal membranes. The E protein is composed of three regions: domains I (EDI), II (EDII), and III (EDIII). EDII preserves the conserved hydrophobic fusion loop required for cell entry, whereas EDIII interacts with cellular receptors. The E protein is a key target for peptides capable of halting the infectivity of flaviviruses [44,45,46], particularly those derived from the C-terminal portion of the E protein [29,33,34,36]. This portion, known as stem, is a membrane-proximal region that plays important roles in the viral fusion process [47,48]. Since the stem region is conserved among flaviviruses [32], stem-based antiviral peptides are potential candidates for cross-inhibiting the infectivity of flaviviruses.

Here we showed that the DV2 stem-derived peptide (aa 419–447), previously reported to inhibit the four DENV serotypes [33], was capable of inhibiting ZIKV infection in cell culture and protected mice against an otherwise lethal challenge with the virus. Interestingly, despite the DV2 peptide being based on the DENV sequence, our in silico docking analyses showed that N- and C-terminal DV2 residues interact with ZIKV EDI and EDII in postfusion conformations. Moreover, this interaction is mainly based on residues from the conserved region (419–441) among the viruses. Accordingly, a two-step mechanism was initially proposed for the DV2-inhibitory activity against DENV infection. First, DV2 binds nonspecifically to the viral lipid layers using C-terminal hydrophobic residues, and, once in the endosome, it specifically interacts with the postfusion form of the E protein (N-terminal residues) and blocks viral endosome membrane fusion [33]. Thus, we assume that the ZIKV cross-inhibition effects observed in the presence of the DV2 peptide are similar to those reported for DENV. 

The DV2 peptide was able to inhibit in vitro ZIKV infectivity by more than 95% at 6–12 μM. A previous study demonstrated that the DV2 peptide inhibited the four dengue types at different molar concentrations, DENV1, IC_90_ = 0.1 μM; DENV2, IC_90_ = 0.3 μM; DENV3, IC_90_ = 2 μM; and DENV4, IC_90_ = 0.7 μM. In contrast, peptides with sequences derived from other flaviviruses (YFV and WNV) did not inhibit DENV infection [32]. Here, the IC_50_ (1.98–2.65 μM) or IC_90_ (2.23–2.7 μM) values established for ZIKV inhibition were higher than those reported for the DENV types. However, IC parameters are difficult to compare as these have been shown to be dependent on the cell line and virus strain. Interestingly, despite the limitations mentioned above, the IC_90_ = 2.7 μM value shown here is close to the IC_90_ found by the authors for DENV3 inhibition, whose stem region has the highest similarity with that of ZIKV (76%) compared with the similarities observed with the stem regions of the other serotypes, DENV1 (52%), DENV2 (55%), and DENV4 (62%). Thus, our in vitro results demonstrate the cross-antiviral activity of a DENV2-derived peptide with ZIKV, highlighting the potential application of stem-derived peptides for the control of different flavivirus infections.

The cross-inhibitory activity of flavivirus stem-derived peptides has also been reported previously. In this context, a 33-residue peptide mimetic to the DENV2 sequence, named DN59 (412–444), was effective against the four DENV serotypes and WNV (IC_50_ = 2–5 μM). The authors suggested that DN59 interacts with the viral lipid membrane, disrupting the lipid bilayer and forming holes from where the viral RNA exits, irreversibly inactivating the virions [29]. Similarly, a ZIKV peptide, designated Z2 (aa 421–453), was shown to disrupt the ZIKV envelope membrane integrity, inhibiting different ZIKV strains, such as SZ01 (IC_50_ = 2.6 μM), FLR (IC_50_ = 4.0 μM), and MR766 (IC_50_ = 13.9 μM); Z2 also inhibited DENV2 (IC_50_ = 4.0 μM) and YFV 17D (IC_50_ = 5.0 μM) [36]. Another study described a stem-derived P5 peptide that inhibited JEV as well as ZIKV infections (IC_50_ = 3.27 μM). The authors also showed that the equivalent peptide ZP1 (424–445) derived from ZIKV inhibited ZIKV infection at lower concentrations (IC_50_ = 1.32 μM) than the JEV-derived peptide [34]. Thus, our data are in line with the literature and support the cross-inhibition effects of E protein stem-based peptides against different flaviviruses, raising perspectives for the development of anti-flavivirus therapies based on the use of synthetic peptides. 

Some stem-based peptides have shown in vivo antiviral effects in animal models [34,36]. Here, we used a highly susceptible ZIKV infection animal model, IFNα/βR^−^/^−^AG129 mice, to evaluate in vivo DV2-induced protection against ZIKV [49]. Our results showed an important DV2-mediated protective effect against ZIKV infection based on different in vivo parameters, including viremia, morbidity, and mortality. Notably, 12 µM of the DV2 peptide protected most (80%) of the animals from death and prevented weight loss. To our knowledge, this is the first study to describe the in vivo antiviral effects of the DV2 peptide. In vivo anti-ZIKV protection has also been reported for the JEV E protein stem-derived peptide P5 [34]. The P5 peptide reduced viral load and histopathological damage in the brains of mice inoculated with ZIKV [34]. Alternatively, the Z2 peptide, derived from the ZIKV E protein stem region, conferred significant protection (75%) against death and neurologic symptoms among challenged mice [36]. Collectively, different synthetic peptides derived from the stem region of the E protein of different flaviviruses represent a potential therapeutic strategy against infection and, in the case of ZIKV, may avoid or reduce neurological damage in newborns associated with infection in pregnant women.

## Figures and Tables

**Figure 1 viruses-15-00839-f001:**
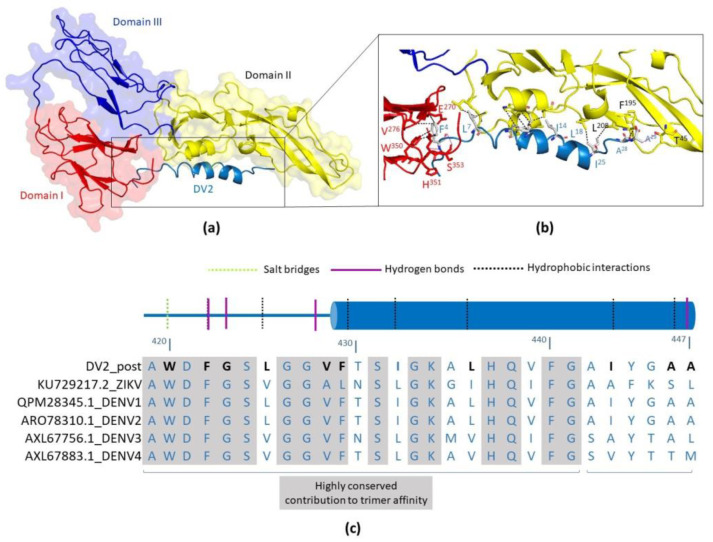
Prediction of DV2 peptide and ZIKV E protein interactions based on in silico docking analyses. The structural model of ZIKV E protein in postfusion conformation (based on the structural coordinates from PDBs 3G7T and 1OK8) was used for the analysis. The protein is shown in surface mode highlighting domain I, II, and III, displayed as red, yellow, and blue colored cartoons, respectively. The DV2 peptide is represented using a sky-blue-colored cartoon. (**a**) According to docking results, the peptide interacts with EDI and EDII, in the postfusion conformation. (**b**) Detailed view of the interacting residues between the ZIKV E protein and the DV2 peptide, predicted by the PLIP program. Residues are shown as colored sticks; sky blue has been used for the DV2 peptide, salmon for DI, and yellow for DII residues. (**c**) Representation of residues 420–447 of the DV2 peptide and their interactions with the ZIKV E protein. The amino acid sequence alignment of the DV2 peptide with the respective sequences of the envelope protein of ZIKV and DENV (types 1 to 4) is shown below. The residues of the peptide that interact with the postfusion conformation are highlighted in bold black.

**Figure 2 viruses-15-00839-f002:**
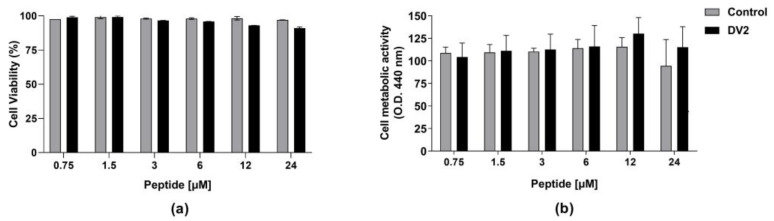
Cytotoxic effects of the DV2 peptide on eukaryotic cells. Vero CCL-81 cell monolayers were treated with increasing concentrations (0.75–24 µM) of the DV2 peptide for up to 24 h. Cell samples were exposed to the peptide solvent (DMSO) at the same volume/dilution (control). (**a**) Percentage of live cells obtained by flow cytometry analyses. (**b**) Cell proliferation measured by metabolic activity using the WST-1 reagent. Data are presented as mean ± SEM of live cells normalized to the respective control.

**Figure 3 viruses-15-00839-f003:**
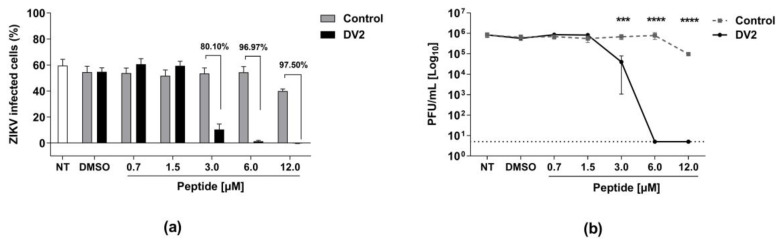
The DV2 peptide inhibits ZIKV infectivity in vitro cultured Vero cells. Vero cell monolayers were infected with ZIKV (MOI = 1.0) in the presence of the DV2 (DV2) or Ph22 (Control) peptides. The impact of the treatment with the DV2 peptide on virus infection was measured at 24 hpi. (**a**) Percentages of infected cells determined by flow cytometry. (**b**) Viable ZIKV titers (PFU/mL) determined by plaque assay using culture supernatants of infected cells. Cells incubated with viruses treated with Ph22, DMSO, or culture medium (NT) were used as negative controls. The data are presented as mean ± SEM (bars) from three independent experiments. Statistically significant differences were determined using two-way ANOVA with Bonferroni correction (*** *p* < 0.001 and **** *p* < 0.0001, compared to the DMSO control).

**Figure 4 viruses-15-00839-f004:**
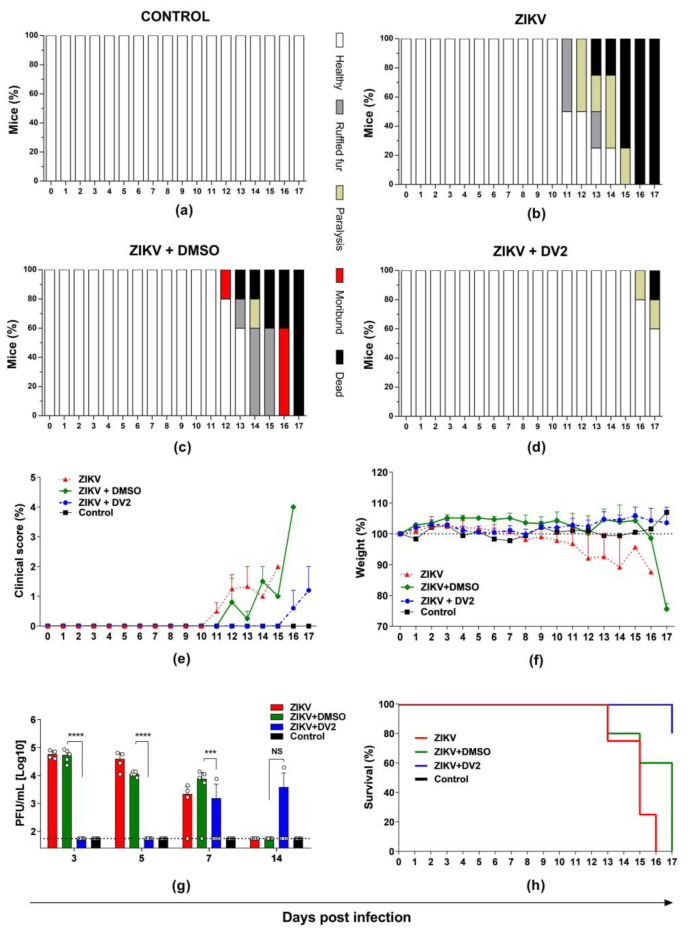
The DV2 peptide induces protection against ZIKV infection in AG129 mice. DV2 (12 µM) was incubated with 100 PFU ZIKV^BR^ for 30 min at 37 °C. The virus-peptide mixtures were s.c. inoculated in AG129 mice. The (**a**–**d**) morbidity, (**e**) clinical score, (**f**) weight changes, (**g**) viremia, and (**h**) survival were monitored in the infected animals for up to 17 days. The data are presented as means ±SD of the respective parameters in the different mice groups. Statistical significance was evaluated using two-way ANOVA followed by Bonferroni correction. Mantel-Cox test was used to analyze the survival results (*** *p* < 0.001; **** *p* < 0.0001; NS, not significant).

**Table 1 viruses-15-00839-t001:** DV2 peptide concentrations required to inhibit ZIKV infection by 50% (IC_50_) or 90% (IC_90_), determined by different methods.

	IC50 (μM)	IC90 (μM)
Flow Cytometry ^1^	2.647	3.228
Plaque assay ^2^	1.976	2.696

^1^ ZIKV infected cells (%), ^2^ Viable ZIKV titers (PFU/mL).

## Data Availability

Not applicable.

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
