# Peer review of "The Anti-Dengue Virus Peptide DV2 Inhibits Zika Virus Both In Vitro and In Vivo"

_viruses, 2023, doi:10.3390/v15040839_

Round 1
Reviewer 1 Report
This manuscript describes a set of experiments designed to test whether a peptide called DV2 that has been shown to have inhibitory effects on dengue infection can also inhibit Zika virus infection. The study was well done for the most part, and the manuscript is very well written and easy to understand. I have the following specific comments:
1. Lines 25-26, the ICTV recommends that virus names should not be capitalized unless they are a proper noun. Thus the correct way to write these names is: Zika virus, dengue virus, yellow fever virus, West Nile virus, Japanese encephalitis virus.
2. The figure 1 legend mentions that some residues in the peptide sequence are in bold black lettering, but I do not see any letters in bold black.
3. As I understand it, the WST-1 assay that the authors employed to test cell cytotoxicity is in fact not a cytotoxicity test. It is a cell proliferation (DNA replication) assay. Thus the authors are measuring effects on DNA replication, not effects on cell viability. There could be cell death occurring in this experiment that would not be evident in this assay, for example if the cell monolayers were confluent and no longer actively dividing. The authors should repeat this experiment with an assay that actually measures cell viability, to ensure that the peptide does not have cytotoxic effects.
4. The flow cytometry assay in Fig 3a is not directly measuring virus replication, it is measuring the percentage of infected cells. It is more of a measurement of virus infectivity than replication. The authors need to revise the text describing this experiment on Lines 195-6 and elsewhere if needed.
5. I could not find any mention of the time points that were sampled to measure virus infectivity in Fig 3a or virus replication in Fig 3b.
Author Response
Response to Reviewer 1 Comments
Point 1: Lines 25-26, the ICTV recommends that virus names should not be capitalized unless they are a proper noun. Thus, the correct way to write these names is: Zika virus, dengue virus, yellow fever virus, West Nile virus, Japanese encephalitis virus.
Response 1: Thank you for your advice. We have indicated the correct virus names in the revised version of the manuscript (lines 27−29).
Point 2: The figure 1 legend mentions that some residues in the peptide sequence are in bold black lettering, but I do not see any letters in bold black.
Response 2: We apologize for the mistake. The figure 1c was edited and those residues are in bold black.
Point 3: As I understand it, the WST-1 assay that the authors employed to test cell cytotoxicity is in fact not a cytotoxicity test. It is a cell proliferation (DNA replication) assay. Thus, the authors are measuring effects on DNA replication, not effects on cell viability. There could be cell death occurring in this experiment that would not be evident in this assay, for example if the cell monolayers were confluent and no longer actively dividing. The authors should repeat this experiment with an assay that actually measures cell viability, to ensure that the peptide does not have cytotoxic effects.
Response 3: Cell metabolic activity assays, such as WST or MTT, are commonly used to determine cell viability (proliferation). However, we agree that metabolic changes do not indicate the cytotoxic effect of the tested drug. Therefore, we performed another assay to determine the number of live cells by staining them with a live/dead fluorescent dye after 24h-incubation with DV2 peptide. The results are shown in Figure 2a and the methods in the heading 2.4 of the revised manuscript.
Point 4: The flow cytometry assay in Fig 3a is not directly measuring virus replication, it is measuring the percentage of infected cells. It is more of a measurement of virus infectivity than replication. The authors need to revise the text describing this experiment on Lines 195-6 and elsewhere if needed.
Response 4: Thank you for reminding us the correct meaning of the methodologies we have employed to show DV2 antiviral activity. Indeed, agree the measures were related to virus infectivity. We have overwritten the term replication by infectivity in the revised manuscript (lines 20, 23, 66, 71, 73, 79, 234, 242, 283, 287, 301).
Point 5: I could not find any mention of the time points that were sampled to measure virus infectivity in Fig 3a or virus replication in Fig 3b.
Response 5: The ZIKV infectivity and replication were measured in one time point (24h after infection) as mentioned in Figure 3 legend (The impact of the treatment with the DV2 peptide on virus infection was measured at 24 hpi).

Reviewer 2 Report
Abstract: Background and Aim of the study is missing. Abstract is starting directly from the methodology.
Introduction is short.
There is a need to add about the global burden of Zika virus and Dengue virus, mortality from these viruses, drug and vaccination efforts etc.
What is the situation of Dengue and Zika in Brazil, Add a paragraph.
Heading 2.1 Ethics statement. line 58-60, is about animal facility, it should be separate from the Ethics Statement heading.
Line 103. What is ZIKVBR? Please explain
heading 2.7: Please explain, how many mice were inoculated with peptide and what was the number of mice in control group.
Figure 2: Control should be DMSO, not another peptide. or at least there should be a single bar of control with only DMSO.
Figure 3: DMSO controls are well shown. if possible add such control in figure 2.
Figure 4: Results of study are very good as shown in figure 4.
Discussion: There is a need to add more references in the discussion.
References are not up to date. Alteast 70% of references should be from the last 5 years.
Line 342. Institutional review board statement is written not applicable.
Add full details of the ethical approval including name of committee who approved study, letter number and date of approval.
Over all study is very interesting and should be published. But there are many minor mistakes. I will suggest major revision to improve the quality of the paper.
Author Response
Response to Reviewer 2 Comments
Point 1: Abstract: Background and Aim of the study is missing. Abstract is starting directly from the methodology.
Response 1: We really appreciate your comment. We have incorporated the background and the study aims in the Abstract of the revised manuscript (lines 12, 13, 14).
Point 2: Introduction is short.
- There is a need to add about the global burden of Zika virus and Dengue virus, mortality from these viruses, drug and vaccination efforts etc.
- What is the situation of Dengue and Zika in Brazil, Add a paragraph.
Response 2: We are grateful for this pertinent suggestion. We expanded the introduction session as suggested and included the additional information in the revised manuscript (lines 29−60).
Point 3: Heading 2.1 Ethics statement. line 58-60, is about animal facility, it should be separate from the Ethics Statement heading.
Response 3: As suggested, we moved the animal facility information from Heading 2.1 (Ethics statement) to Heading 2.8 (Mice infection) (lines 149−151).
Point 4: Line 103. What is ZIKVBR? Please explain
Response 4: The ZIKVBR was isolated from a clinical case during the 2015 Brazilian outbreak (isolate="BeH823339'', GenBank: KU729217) (doi:10.1126/science.aaf5036). This information was incorporated in the Heading 2.2 text (lines 91−93).
Point 5: heading 2.7: Please explain, how many mice were inoculated with peptide and what was the number of mice in the control group.
Response 5: We incorporated the number of animals of all experimental groups investigated in our study in Heading 2.8.
Point 6: Figure 2: Control should be DMSO, not another peptide. or at least there should be a single bar of control with only DMSO.
Point 7: Figure 3: DMSO controls are well shown. if possible, add such control in figure 2.
Response 6−7: Indeed, DMSO was used as control in this assay of experiments depicted in Figure 2. We used 2.11 ml of DMSO (the volume used at 24 mM DV2 concentration) and, then, performed serial dilutions of both peptide and DMSO. The peptide Ph22 was used in controls of experiments depicted in Figure 3! It was our mistake to refer it in Figure 2 and the correction was inserted in Heading 2.4 (line 108), Heading 3.2 (line 221), and in Figure 2 legend (line 226).
Point 8: Figure 4: Results of study are very good as shown in figure 4.
Response 8: We really appreciate your comment.
Point 9: Discussion: There is a need to add more references in the discussion.
Point 10: References are not up to date. Alteast 70% of references should be from the last 5 years.
Response 9−10: We attempted to discuss about “stem”-derived peptides with antiviral activity against Flavivirus to take the reader’s attention to this particular region of the E protein. Since there are not recent published studies specifically targeting the stem region, we included other recent references about antiviral peptides that interact with the E protein (line 283).
Point 11: Line 342. Institutional review board statement is written not applicable.
Response 11: We apologize for the missing information. We have added such statement in the revised manuscript (lines 367−370).
Point 12: Add full details of the ethical approval including name of committee who approved study, letter number and date of approval.
Response 12: Thank you for your observation. The name of committee who approved study, letter number and date of approval are indicated in the Heading 2.1.
Point 13: Overall study is very interesting and should be published. But there are many minor mistakes. I will suggest major revisions to improve the quality of the paper.
Response 13: We are thankful for your pertinent suggestions.

Reviewer 3 Report
The authors applied DV2, an anti-DENV synthetic peptide, to the scenario of ZIKV infection, and found that DV2 also exerts efficient anti-ZIKV activity both in vitro and in vivo. This manuscript has provided concise but consistent data showing that beside DENV, DV2 could also be considered as a promising therapeutic candidate curbing the replication of ZIKV. Overall, the experiments were well-designed, and results were presented clearly and straightforwardly. The following comments I listed should be clarified.
1 In Fig 1c, the AQU11806.1 peptide used for alignment was from an African strain, while a Brazilian strain KU729217.2 was used for experiments. Please provide the evidence showing that these two strains share the conserved DV2 binding sequence in envelope.
2 What did the dotted line, located around 80%, indicate in Fig 2?
3 The ZIKV dose used in mice was 100 PFU shown in line 241, while in line 121 it was 102 PFU. Please make them consistent and accurate.
Author Response
Response to Reviewer 3 Comments
Point 1: In Fig 1c, the AQU11806.1 peptide used for alignment was from an African strain, while a Brazilian strain KU729217.2 was used for experiments. Please provide the evidence showing that these two strains share the conserved DV2 binding sequence in envelope.
Response 1: Thank you for indicating this mistake! We have replaced the African strain by the Brazilian ZIKV strain sequence in the alignment (Figure 1c). We also verified that similarities among ZIKV (KU729217.2) and DENV1-4 subtypes related to the DV2 peptide stem region were the same as the similarities we had found using the ZIKV (AWW21436) (lines 311−312).
Point 2: What did the dotted line, located around 80%, indicate in Fig 2?
Response 2: We intended to demonstrate that cell viability was greater than 80% but we haven't mentioned it in the text. The dotted line was removed from Figure 2 in the revised version of the manuscript.
Point 3: The ZIKV dose used in mice was 100 PFU shown in line 241, while in line 121 it was 102 PFU. Please make them consistent and accurate.
Response 3: Thank you for this observation. We have opted for 100 (instead of 102) PFU (lines 151, 271).

Round 2
Reviewer 1 Report
The authors have addressed my comments. The only thing that wasn't corrected is the word "fever" in yellow fever virus is still incorrectly capitalized on line 28.
Author Response
Point 1: the word "fever" in yellow fever virus is still incorrectly capitalized on line 28.
Response 1: Thank you again! We have replaced “Fever” to “fever” (line 28).
Reviewer 2 Report
Manuscript is improved after revision. Accepted in current form.
Author Response
Point 1: Manuscript is improved after revision. Accepted in current form
Response 1: We really appreciate your suggestions!